# How GBS Got Its Hump: Genomic Analysis of Group B *Streptococcus* from Camels Identifies Host Restriction as well as Mobile Genetic Elements Shared across Hosts and Pathogens

**DOI:** 10.3390/pathogens11091025

**Published:** 2022-09-08

**Authors:** Chiara Crestani, Dinah Seligsohn, Taya L. Forde, Ruth N. Zadoks

**Affiliations:** 1Department of Global Health, Institut Pasteur, 75015 Paris, France; 2Institute of Biodiversity, Animal Health and Comparative Medicine, University of Glasgow, Garscube Campus, Glasgow G61 1AF, UK; 3National Veterinary Institute, SE-751 89 Uppsala, Sweden; 4Sydney School of Veterinary Science, Faculty of Science, The University of Sydney, Camden, NSW 2570, Australia

**Keywords:** group B *Streptococcus*, *Streptococcus agalactiae*, camel, milk, GWAS, host adaptation

## Abstract

Group B *Streptococcus* (GBS) literature largely focuses on humans and neonatal disease, but GBS also affects numerous animals, with significant impacts on health and productivity. Spill-over events occur between humans and animals and may be followed by amplification and evolutionary adaptation in the new niche, including changes in the core or accessory genome content. Here, we describe GBS from one-humped camels (*Camelus dromedarius*), a relatively poorly studied GBS host of increasing importance for food security in arid regions. Genomic analysis shows that virtually all GBS from camels in East Africa belong to a monophyletic clade, sublineage (SL)609. Capsular types IV and VI, including a new variant of type IV, were over-represented compared to other host species. Two genomic islands with signatures of mobile elements contained most camel-associated genes, including genes for metal and carbohydrate utilisation. Lactose fermentation genes were associated with milk isolates, albeit at lower prevalence in camel than bovine GBS. The presence of a phage with high identity to *Streptococcus pneumoniae* and *Streptococcus suis* suggests lateral gene transfer between GBS and bacterial species that have not been described in camels. The evolution of camel GBS appears to combine host restriction with the sharing of accessory genome content across pathogen and host species.

## 1. Introduction

Group B *Streptococcus* (GBS), or *Streptococcus agalactiae*, is primarily known as a cause of human neonatal and maternal disease [1]. However, it is a versatile multi-host pathogen that also inflicts significant damage in adults [2,3,4] and in several animal species [5,6,7]. The different host species can be seen as different niches for GBS, whilst niches can also be differentiated within host species. For example, hypervirulent clonal complex (CC) 17 is a human GBS clade specifically associated with neonates. In cattle, GBS is specifically associated with the mammary gland, where it causes infection and inflammation (mastitis). The ability of bacterial pathogens to thrive in different niches can be driven by adaptation of the core genome, such as point mutations in genes of particular functional relevance, or changes in accessory genome content [8,9]. Host-adapted lineages of GBS, such as CC17 in humans, CC61/67 in cattle and CC552 in poikilothermic species, are identified based on multilocus sequence typing (MLST), which represents the core genome. However, the accessory genome appears to be particularly important in GBS: the ability of host generalist lineages such as CC23 (humans and cattle) or CC283 (humans and fishes) to infect multiple host species is enabled by the acquisition of host-associated accessory genome content [6,7,10]. This occurs predominantly through the horizontal gene transfer (HGT) of mobile genetic elements (MGE), including transposons, plasmids, phages, genomic islands (GEI), and phage inducible chromosomal islands (PICI) [11,12]. The ensemble of MGE carried by a given species, also known as the mobilome [13], plays a fundamental role in shaping the adaptability of bacteria, and may affect the host range as well as tissue tropism [14].

The colonisation or infection of a new niche provides access to new accessory genetic material and creates opportunities for the acquisition of niche-associated genes that may facilitate amplification and subsequent onward transmission [7]. This is postulated to have happened to GBS CC283, which expanded in fishes around the time of the intensification of aquaculture, and might be explained by initial human-to-animal transfer with the subsequent acquisition of fish-associated MGE that allowed for amplification in the new niche [3,6]. Likewise, the acquisition of a lactose operon is thought to explain the expansion of CC1 and CC103 in cattle [15]. The spill-over of GBS between host species happens in multiple directions, including between its three major host groups, i.e., humans, cattle, and fishes, which contain different microbiomes and hence provide an opportunity for exposure to and acquisition of a diversity of MGE [7]. Thus, the GBS pangenome and its evolution can only be understood by studying its ecology and genomics across host species. Increasingly, this is being done for GBS in humans, cattle, and fishes [16,17], but an important emerging host species has received limited attention: the one-humped camel (*Camelus dromedarius*).

GBS has been reported in camels in East Africa [18,19,20] and the Middle East [21]. In camels, as in humans, GBS is both a commensal and a pathogen, with carriage in close to a quarter of the camel population. Carriage in camels occurs primarily in the nasopharynx [22,23,24,25], in contrast to the situation in humans, where the gastrointestinal and urogenital tract are the most common carriage sites. As in dairy cattle, GBS in camels is commonly associated with mastitis. In camels, however, GBS is not limited to the mammary gland and may also cause skin and soft tissue infections [18]. The distinct ecology of camel GBS could be expected to reflect evolutionary adaptations that are unique to camels. Based on limited genomic data and multi-locus sequence typing, camel GBS appears to be distinct from GBS of humans or other animals [19,26], except for one report in the camel milk of a sequence type (ST) 1 isolate that showed genetic features of adaptation to the human host, pointing to a possible human-to-camel transmission event [25,27]. Analysis of a single camel-derived GBS genome suggested distinct gene content compared to GBS from other host species, and biochemical enrichment of carbohydrate metabolism [7]. As camel husbandry in East Africa is on the cusp of intensification [25], it is quite possible that new strains of GBS in camels will emerge in this setting. Thus, now is an opportune time to study the adaptation of GBS to a hitherto poorly represented host species, including different organ systems within this host species, and to explore the potential for HGT between GBS clades or spill-over of GBS strains into or out of the camel population.

To gain insight into potential genetic mechanisms for the host adaptation of GBS in camels, we investigated the population structure of camel GBS. To identify host-associated core and accessory genome content, camel GBS was compared to representative GBS genomes from major host groups and lineages. Because commonly used typing schemes for GBS have largely been developed for human isolates, they may miss animal-specific genome content. Therefore, we used genome-wide association studies (GWAS), which are an unbiased statistical approach to detect genes that are positively or negatively associated with a trait [28]. In addition to host as a trait, we considered mastitis as a trait because of the importance of this disease in terms of camel health and welfare, as well as the contribution of camels to nutrition security in the Horn of Africa.

## 2. Methods

### 2.1. Dataset Selection

A total of 680 GBS genomes were used for this study, including all publicly available camel GBS genomes (*n* = 139), and 541 GBS genomes from the three major GBS host groups for comparison (human, *n* = 285; bovine, *n* = 155; piscine, *n* = 101). The latter were selected to represent all major sublineages (SL) and clonal groups (CG), and a wide range of geographical and temporal origins (35 countries; 1953–2019). We use the terms SL and CG to refer to phylogenetic clusters that include several sequence types (ST), with the latter being at a finer scale of resolution. To avoid redundancy in the genomic data, which would have increased computational requirements without providing additional information on gene presence/absence, we selected one isolate per host/year/serotype/ST combination for each epidemiological unit, e.g., hospital or farm in the comparator dataset. Camel GBS originated from six sample types or sampling sites (mammary gland/milk, *n* = 81; nose, *n* = 38; oropharynx, *n* = 8; abscesses, *n* = 6; rectum, *n* = 3; skin/wounds, *n* = 3). A complete list of camel and GBS genomes from other species used in this study can be found in the Appendix A. Genomes originating from public repositories (*n* = 633) were downloaded in the form of raw reads (either paired-end or single-end) when possible (*n* = 462), or in the form of assembled genomes when raw data were not available (*n* = 171). Data were newly generated for human (*n* = 4), bovine (*n* = 1) and piscine (*n* = 42) GBS isolates from Thailand (*n* = 33) and Vietnam (*n* = 14) by Prof. Swaine Chen (Genome Institute of Singapore, Singapore) and used with permission from Dr. Wanna Sirimanapong (Mahidol University, Nakhon Pathom, Thailand) and Dr. Nguyen Ngoc Phuoc (Hue University of Agriculture and Forestry, Hue City, Vietnam), respectively.

### 2.2. Sequencing, Assembly and Typing

The sequencing of the 47 new genomes was carried out with Illumina HiSeq technology at MicrobesNG (Birmingham, UK) (*n* = 11) and at the Genome Institute of Singapore (*n* = 36). Sequence data have been made publicly available on ENA (PRJEB53664). *De novo* assembly of reads was conducted for 509 genomes. The Wellcome Sanger Institute assembly pipeline with velvet v1.2.10 was used for genomes that were already present within the Sanger server (*n* = 159) and SPAdes v3.13.0 for the remaining genomes with available raw reads (*n* = 350, including the 47 newly sequenced isolates). The assembly quality for all 680 genomes was checked with QUAST v5.0.2 [29] (Appendix A), and species identity was confirmed with KmerFinder v3.2 [30].

Seven-gene multi locus sequence typing (MLST) was carried out for all 680 genomes, using SRST2 v0.2.0 [31] on raw reads or MLST v2.0 [32] when only assemblies were available. The capsular type was detected *in silico* using a BLAST-based standard method [33], which was developed for human GBS and previously validated for bovine GBS [15]. To test this method in camels, we compared our results with previously reported capsular types obtained with a different method [34] when these data were available (*n* = 17) [18]. For the entire dataset, we compared capsular types obtained with the method from Metcalf et al. [33] with those obtained using a new tool for capsular typing, GBS-SBG [35], which allows for differentiation of type III subtypes (1–4). For camel GBS only, capsular type loci (*cps*) were extracted from the genomes with *in silico* PCR within SnapGene v6.0.2 (from Insightful Science; available at snapgene.com) (forward primer: *cps*-F 5${}^{\prime }$-ATGTCTAATCATTCGCGCCGTCAACAAAAGAAA-3${}^{\prime }$; reverse primer: *cps*-R 5${}^{\prime }$-TTATAAGGTTTTAACTTCGTCTACAAATAATTG-3${}^{\prime }$). When *cps* loci were found separated into two contigs (*n* = 18), the sequences were manually extracted, and the *cps* locus was scaffolded by adding a series of 100 ‘Ns’ between the two segments. Pseudogenisation of the capsular genes was checked manually by comparing the length and number of the open reading frames (ORFs) (as predicted by SnapGene) among sequences belonging to the same *cps* type. Prokka v1.14.5 [36] was used for genome annotation.

### 2.3. Core Genome Analysis of Camel Isolates

A phylogenetic tree of the whole dataset was estimated from the core gene alignment obtained from Roary (as described in the next section) with IQtree v1.6.10 [37], with a general time-reversible (GTR)+G model. Fastbaps v1.0.4 (fast hierarchical Bayesian analysis of population structure) [38] was run on the core genome alignment of camel isolates belonging to SL609 within RStudio v1.3.1093 [39], R v4.0.3 [40].

### 2.4. Genome-Wide Association Studies (GWAS)

Two pan-GWAS were performed with Scoary v1.6.16 [28]: (i) markers of host association were identified through comparison of camel against non-camel GBS genomes; (ii) markers of mastitis-association within camels were identified through comparison between GBS from camel milk (obtained from mammary glands with mastitis) and extra-mammary camel body sites or fluids (including nose, oropharynx, abscesses, rectum and skin/wounds) [25,27]. Gene presence/absence matrices were generated with Roary v3.13.0 [41] from annotation files created with Prokka. For each genetic variant, the Scoary output includes *p*-values, sensitivity (SE) and specificity (SP) measures. Sensitivity indicates which proportion of isolates from the niche of interest (here: camel or milk) contains the specified feature. Specificity indicates which proportion of isolates from other niches (here: other hosts, or other sample types) does not contain the specified feature. For example, perfect sensitivity and specificity at host level would mean that all camel GBS isolates contain the sequence of interest, but none of the GBS isolates from other hosts do. A flow chart of the methods from sample origin through to GWAS can be found in the Appendix A.

### 2.5. Analysis of Mobile Genetic Elements in Camel Isolates

Presence of prophages and PICI was assessed by screening genomes for integrase gene types with a BLAST-based method, as previously described [12]. Thresholds were set at 90% for prophages and 96% for PICI for percentage of identity (ID), and at 99% for query coverage. Considering that only one camel genome was available when this method was developed, we also screened these assemblies with PHASTER [42] to ensure all prophages and PICI were detected by BLAST. Prophages and PICI which carried an integrase gene detected by PHASTER were classified based on integrase type [12]. We then compared results from the two analyses.

Figures were created with Phandango v1.3.0 [43], Circos v0.69-5 [44], BRIG v0.95 [45], and Easyfig v 2.2.2 [46], and modified with Inkscape v1.2.1 (www.inkscape.org).

## 3. Results

### 3.1. Genotyping and Serotyping

Our dataset included 113 ST belonging to 15 sublineages (see Appendix A). All camel GBS belonged to SL609 (Figure 1), with the exception of a single ST1 isolate [27]. SL609, which did not include any GBS genomes from other host species, comprised 12 ST (see Appendix A) and capsular types II–VI (Figure 1). Within SL609, fastbaps identified five subpopulations (Figure 1). These largely corresponded to clonal groups (CG), e.g., fastbaps population 2 corresponds to CG615 (including one ST612 and all ST615, all with capsular type II except for one type III) and fastbaps population 5 corresponds to CG616 (including ST616, ST1653 and ST1654, all with capsular type III).

Across the entire dataset, results obtained with the two *cps* typing methods largely agreed, but there were some inconsistencies and ambiguities. Among camel isolates, one ST615 had capsular type III based on the method developed by Metcalf et al. [33], but type II or possibly type IV (strong second best match) based on the GBS-SBG method from Tiruvayipati et al. [35]. This *cps* locus showed pseudogenisation (see below); in particular, there was an assembly gap upstream from the gene *cpsK*, possibly due to the presence of MGE-associated repeated sequences. ST616 isolates had capsular type III based on the Metcalf method, with nearly identical matches for subtypes III-4 and III-1 (Bit Score: 0.989 and 0.997, respectively) based on GBS-SBG. Finally, three isolates gave one best match with the first method, but two possible matches with the second (ST612 type IV vs. IV/V; ST614 type V vs. V/IX; ST617 VI vs. VI/III-1) (see Appendix A). Discrepancies were also observed between results from the two methods used in the current study, and those previously published for the same genomes: several isolates with serotype IV based on our analyses had previously been reported as capsular type Ia based on a multiplex PCR method [18,34]. In addition, an isolate with capsular type III based on our analyses was previously reported as type VI (ILRI021) with the same band-based method [18]. To explore the potential origin of those differences, the (*cps*) sequences of all camel capsular type IV isolates were aligned with a reference sequence from human GBS type IV (LT671987.1) using Geneious Prime v2021.1 [47], revealing high SNP density in the region straddling *cpsN* and *cpsJ* (Figure 2). Based on bioinformatic analysis, the SNPs likely prevent binding of one of the primers for *cpsJ*, which is used to distinguish type IV from Ia [34] (Appendix A). We also detected a series of InDels (insertions/deletions) in capsular type IV, leading to pseudogenisation in a minority of isolates (*n* = 4). Pseudogenisation was also observed in other capsular types (type II, *n* = 1; type III, *n* = 27; type V, *n* = 2; type VI, *n* = 3), and involved multiple genes within the capsular operon, in particular *cpsI* (*n* = 13 genomes), and *cpsE* (*n* = 7 genomes). In total, 37 camel GBS genomes showed pseudogenisation of the capsular operon (Appendix A).

### 3.2. Markers of Camel-Association Within GBS

Only genes whose *p*-values reported by Scoary were <0.001 are described here; the exact *p*-values can be found in the Appendix A.

Among the 100 best-scoring genes in the host-level GWAS, 28 were positively and uniquely associated with GBS from camels (camel-specific; SP = 100%) (Appendix A). High-scoring genes largely mapped to two genomic islands (GEI) (Figure 3). GEI1 included genes for carbohydrate metabolism (sugar phosphatase), DNA modification (DNA topoisomerase and DNA cytosine methyltransferase) and protein channels (voltage-gated chloride channel family *eriC*). In GEI2, high-scoring genes included a sensor histidine kinase for copper and silver (*cusS*), IMMA/IrrE family metallo-endopeptidase, and CPBP family intramembrane metalloproteases (highly sensitive and 100% specific for camel GBS, Appendix A), as well as a transcriptional regulator (Rgg/GadR/MutR), an oxidoreductase (Gfo/Idh/MocA), a histidine phosphatase, an N-acetyltransferase (GNAT family), an ATP-dependent DNA helicase and a pyrimidine nucleotidase (YjjG). GEI2 showed lower GC content than the remainder of the GBS genome (GC GEI2: 31% vs. GC whole genome reference HF952106: 35%) (Figure 3). Alternating opposed GC skews were detected in this area of the genome and corresponded to several MGE (Appendix A), including GEI1 (positive GC skew, average GC content 35%), partial prophage GBS*Int*1 immediately followed by partial ICE*Sp*1108 from *Streptococcus pyogenes* [48] (opposed GC skew peaks, GC content 36% for the prophage and 30% for the ICE), Tn*916* (positive GC skew, GC content 38%), ICE*Sp*1108 (negative GC skew, GC content 31%), and GEI2 (positive GC skew, GC content 31%). Other significant hits included a carbohydrate kinase (FGGY family), a PTS sugar transporter subunit IIC, three insertion sequences (IS) (IS*1562*, IS*Lre2* and IS*Sag9*) and a type I restriction modification system (RMS) (subunit M) associated with camel GBS.

### 3.3. Markers of Mastitis-Association within Camel GBS

Genes that were strongly and positively associated with isolates from milk compared to those from other camel samples (Appendix A) included a hyaluronate lyase (*hylB*) variant (Figure 1), genes coding for transporters and enzymes (ABC transporter, NAD-dependent malic enzyme, deacetylase, phosphatidate cytidylyltransferase, epoxyqueuosine reductase), and five *cps* gene variants (*cpsC*, *cpsH*, *cpsI*, *cpsJ* and *cpsK*). These *cps* variants were associated with capsular type III and ST616, which is the predominant serotype/ST combination among GBS isolates from camel milk [27].

Two GEI carrying milk-associated genes were identified (Figure 4): Tn*916*, which carries *tet*(M), known to be associated with milk isolates from camels [20,27], and a 14-gene cluster of ∼15,000 bp that shows signatures of mobility (transposase, relaxase and mobilisation genes) and carries virulence gene *virD4*. Tn*916* was often linked to a long gene cluster (∼43,000 bp, Figure 4A) that was also significantly associated with milk isolates. PHASTER identified this region as an incomplete prophage because it was lacking its integrase gene, indicating that it might be hijacking Tn*916* to be mobilised. When the region surrounding the Tn*916*-prophage combination was blasted against the general ICEberg database [49] with multigene BLAST, it showed high sequence similarity with a segment of Tn*1207.3* from *Streptococcus pyogenes* strains 2812A [50] and MGAS10394 (Figure 4), and with two composite MGE from *Streptococcus suis* (CMGETZ080501 and CMGEYY060816) (Figure 4A). This prophage, named ϕ1207.3 [51], carries a type II toxin/antitoxin system with a Phd/YefM family antitoxin followed by a Doc (death-on-curing) family toxin. Among the genomes that carried Tn*916* (*n* = 101), 12 did not carry ϕ1207.3, whereas only a single genome (ST617) was ϕ1207.3-positive but *tet*(M)/Tn*916*-negative (Appendix A). Accuracy (combined sensitivity and specificity) for milk-associated genes was generally not as high as for host-associated genes (Appendix A). Lac.2 was detected in 70% of milk isolates, but sensitivity of lactose genes as an indicator of the milk-derived phenotype was low (1.2 to 58%) due to the existence of multiple variants of most *lac* genes: (*lacA*, *n* = 3; *lacB*, *n* = 2; *lacC*, *n* = 4; *lacD*, *n* = 3; *lacE*, *n* = 2; *lacF*, *n* = 4; *lacG*, *n* = 3; *lacR*, *n* = 5; *lacX*, *n* = 2). This also explains the comparatively low Scoary rankings for this milk-associated operon (positions 104 *lacF*_1, 150 *lacX*_1, and 243-965 for *lacABCDEG*).

### 3.4. Mobile Genetic Elements in Camel GBS

Based on PHASTER, 16 (11.5%) of the camel GBS genomes carried complete prophages, 51 (36.7%) carried complete and incomplete (remnant) prophages only, 61 (43.9%) carried remnant prophages only, and 11 (7.9%) carried no prophages. No more than two complete prophages were detected per GBS genome. The BLAST-based method identified prophage integrase types in 88 (63.3%) of the camel GBS genomes, including all genomes with complete prophages according to PHASTER, and with up to three hits per genome. The most common integrase types were GBS*Int*11.1 (*n* = 55) and GBS*Int*10 (*n* = 35), followed by GBS*Int*1 (*n* = 15), GBS*Int*6.1 (*n* = 8), GBS*Int*11.2 (*n* = 5) and GBS*Int*5 (*n* = 3). Two new PICI were detected (Appendix A): PICI3 (length = 13,723 bp), which shares its insertion site with PICI1 and PICI2 (*rpsD*—30S ribosomal protein S4), and PICI4 (length = 12,919 bp), which is found between a hypothetical gene and an aminoacyl-tRNA synthetase gene (amino acid sequences for PICI3*Int* and PICI4*Int* can be found in the Appendix A). Sixty-five genomes (46.8%) were positive for at least one PICI integrase. PICI4*Int*, which carries the virulence-associated protein E (*vapE*) was found across all fastbaps camel populations, particularly in population 4 (CG617 and CG1652), which includes multiple serotypes and sample types and comprises 72.4% of the 58 PICI4*Int*-positive camel GBS genomes.

## 4. Discussion

Genomic analysis of GBS in humans, cattle and fishes has shown that events in both the host population and the pathogen population may contribute to niche adaptation: the emergence of GBS in fishes is attributed to the global expansion and intensification of fish farming [3,52] combined with the acquisition of niche-associated MGE, notably locus 3, which includes genes for galactose metabolism [6,7], whilst adaptation of GBS to humans and cattle, respectively, is attributed to the acquisition of tetracycline resistance and the Lac.2 operon encoding lactose metabolism [53,54]. Expansion in a new host population may be followed by host specialisation, as exemplified by CC552, which is restricted to fish and other cold-blooded animals as a result of genome reduction [10], and by bovine-adapted lineage CC61/67, which is undergoing pseudogenisation of its capsule operon, resulting in loss of a key virulence factor for infection of humans [55]. Here, we present the first large-scale genomic study of GBS from camels. Camel husbandry in Africa is undergoing intensification due to expansion of the camel milk market [27], providing a unique opportunity to explore the current population composition and, subsequently, its evolution under changing selection pressures.

We show that the GBS population from camels is largely distinct from GBS found in other host species in terms of its core genome. With the exception of one isolate belonging to ST1 [27], all camel GBS genomes cluster in one SL (Figure 1). Earlier studies had identified two groups of camel GBS isolates based on MLST maximum-likelihood phylogeny [18], but when considering the entirety of the core genome and all known GBS SL from different hosts, camel GBS forms a single monophyletic lineage (SL609). Not only did camel GBS constitute a unique lineage, but it also harboured a unique variant of capsular type IV. The divergence between capsular type IV in GBS from humans and camels affected capsular type prediction in camel isolates based on a PCR method developed for human isolates [34]. The high prevalence of serotype IV and VI among camel GBS, as described previously [25], and the novel variant of serotype IV may provide a reservoir of capsular variants that are not covered by human GBS vaccines that are currently under development. This could potentially lead to the evolution of vaccine escape mutants, similar to what has been observed in *Streptococcus pneumoniae* [56], especially if the transfer of capsular operons between strains—and the transfer of strains between host species—is possible. Although rare, the detection of ST1 in camels [25] and evidence for capsular switching in GBS [57,58,59] suggest that both processes can occur. An example of capsular switching in camel GBS is shown in Figure 1, where a single member of CG615 has capsular type III whilst the remaining eleven isolates in this CG have capsular type II. As in cattle GBS [55], there is evidence of pseudogenisation in camel GBS. This may limit animal-to-human transmission, as the GBS capsule is considered essential for pathogenicity in humans, where it limits complement deposition and phagocytosis and masks surface proteins to avoid stimulating the immune response [60,61]. Pseudogenisation, however, only affects 37% of the bovine-associated SL61/67 and 27% of camel-associated SL609, i.e., the capsular operon is intact in the majority of isolates from both SL so spillover may still be possible. Indeed, the detection of CC61/67 in humans, albeit extremely rare, has already been described in China [62], whereas data from east Africa are too scarce to prove or rule out the possibility of human infection with SL609.

Using GWAS, we identified accessory genome content that is both uniquely associated with camel GBS (SP = 100%) and almost universally present in camel GBS (SE > 98%). As in bovine [54] and piscine [6] GBS, camel-associated accessory genome content was concentrated in GEI. These GEI were found in an area of the genome that shows signatures of multiple integration/recombination events, including alternating opposed GC skew and variation of the GC content. This genomic region also contained a partial ICE from *S. pyogenes*. Likewise, accessory genome content that was significantly associated with isolates from milk based on within-host GWAS included a prophage that has been described as part of other ICE/MGE in *S. pyogenes* and *S. suis*. Although *S. pyogenes* is thought to be an exclusively human pathogen, whereas *S. suis* is primarily associated with pigs [50,51], MGE with high homology to *S. pyogenes* have also been reported in bovine GBS [15]. In addition to ICE, prophages may contribute to HGT. Complete prophages of type GBS10 and GBS11 [12] were common among camel GBS. They were previously described only among human GBS isolates, and in one seal isolate that likely originated from anthropogenic contamination of the aquatic environment [6]. Phages can facilitate DNA exchanges through transduction [63], including lateral transduction, which involves extended areas of the genome [64] and exceeds chromosomal mobility of classical MGE [65]. A third mechanism that may contribute to HGT between GBS across host species are PICIs. Previous work identified a low diversity of PICI in GBS [12] compared to other bacterial species [11], with only two types detected. PICI1 was widespread across isolates from multiple hosts (humans, fish, cattle, a dog and a dolphin), whilst PICI2 was found uniquely in a camel GBS genome from Kenya [12]. In this study, we identified two new PICI in camel GBS. PICI3 had the same integration site as PICI1 and PICI2, the *rpsD* gene, confirming this site is an important hotspot for recombination of PICI in GBS. The second novel PICI, PICI4, carried a virulence-associated protein (*vapE*) that has also been described in *Rhodococcus equi*, *S. suis* and *S. pneumoniae* [66,67,68], reinforcing the notion that pangenome evolution needs to be explored and understood beyond host- and pathogen species level.

Genes that were associated with the camel as a host included those involved in various metabolic processes, e.g., carbohydrate and metal utilisation. The utilisation of substrates that are present in certain niches is a well-described mechanism of bacterial adaptation [14]. In GBS, it manifests at the host level, as illustrated by the galactose genes associated with piscine GBS [6,7] and by sugar phosphatases associated with the camel host, and also at the organ level, as demonstrated by the association of the lactose operon Lac.2 with isolates originating from milk and, hence, the mammary gland. In camel GBS, Lac.2 was found in 70% of milk isolates, compared to 98–100% of bovine milk isolates [2,15], and the presence of Lac.2 is not exclusive to milk isolates in camels. This suggests that Lac.2 is not necessary for GBS to successfully colonise and establish infections in the udder of camels, but that it offers an evolutionary advantage in adaptation to the mammary gland, similar to the situation reported for *Klebsiella* spp. and *E. coli* [69,70]. The role of the *cusS* gene, normally involved in sensing copper and silver, is unclear. The ability to uptake and utilise essential metals can counteract the host immune defences, which usually sequester these molecules to protect the host from infection [71], but *cusS* has also been described as a contributor to metal homeostasis or metal resistance, notably in *Klebsiella* spp. and *E. coli* [72,73]. In camels, there is a third albeit speculative possibility: copper levels in camel plasma have been reported to be lower than those in bovine plasma [74]. Thus, there may be an advantage to having the ability to acquire a scarce resource that supports cellular processes.

Another milk-associated gene was a gene variant of *hylB*, whose product is able to degrade components of the extracellular matrix. Although *hylB* is believed to contribute significantly to invasion [61,75], hyaluronidase activity is not essential for human GBS infections [76], and GBS associated with neonatal disease may show a disrupted *hylB* gene [77]. To our knowledge, there is no evidence of the functional relevance of *hylB* in the context of milk or mastitis. Considering the population structure within the camel-associated GBS lineage, the *hylB* variant may show a statistical association with camel milk through its association with CG616. Transposon Tn*916* was linked to an incomplete prophage (ϕ1207.3), and both were significantly associated with camel milk in the within-host GWAS. Unlike the element described in *S. pyogenes*, ϕ1207.3 in GBS did not carry macrolide resistance genes. In our experience, camel owners in East Africa mostly use tetracycline, penicillin or streptomycin, whereas we are not aware of macrolide use or a potential selective advantage of macrolide resistance in camels. The incomplete prophage from camel GBS did encode the same toxin/antitoxin (TA) system (Phd/Doc) as in *S. pyogenes*. Phd/Doc TA systems have been described in various gram-positive [78,79] and gram-negative bacterial species [80,81]. Although the biological impact of TA systems in virulence and pathogenicity remains unclear [79], they seem to confer an advantage under stressful environmental conditions for the bacteria [79,82]. Finally, a gene cluster that was associated with camel milk and CG616 carried the *virD4* gene, a previously identified type IV secretion system (T4SS) [83]. This gene was recently described in GBS as significantly associated with CC19 [17], a lineage primarily associated with human infections. It is also found in other bacterial species, including *S. suis* [84], and it is associated with conjugation, the translocation of virulence factors [85,86], anti-phagocytic activity and a pro-inflammatory effect [84].

Although our study was limited geographically (all GBS genomes from camels available to date originated from Kenya and Somalia) and could not elucidate the functional relevance of camel- or milk-associated genes identified by GWAS, we clearly demonstrate the existence of a GBS lineage (SL609) that is both fully specific to camels, and encompasses all but one known camel GBS isolates. Although the camel GBS core genome is phylogenetically distinct, we identify commonalities in the accessory genome of human and camel GBS, e.g., Tn*916* and shared prophage integrase sites, and of bovine and camel GBS, e.g., Lac.2. Moreover, we identified multiple MGE, e.g., ICE*Sp*1108 and prophage ϕ1207.3, that show high sequence similarity to MGE from other streptococcal species, including those that are largely limited to other host species, notably humans or pigs. Thus, camel GBS seems to have an expansive mobilome in terms of host and pathogen species of origin. As camel husbandry and milk production intensify in light of climate change, the animal’s drought resilience, and the growing demand for camel milk, the emergence and expansion of new GBS strains in or derived from camels appears to be a distinct possibility. Ongoing surveillance of the prevalence and population composition of GBS in camels is recommended so that the emergence of resistant, virulent, or zoonotic strains can be detected early. 

## Figures and Tables

**Figure 1 pathogens-11-01025-f001:**
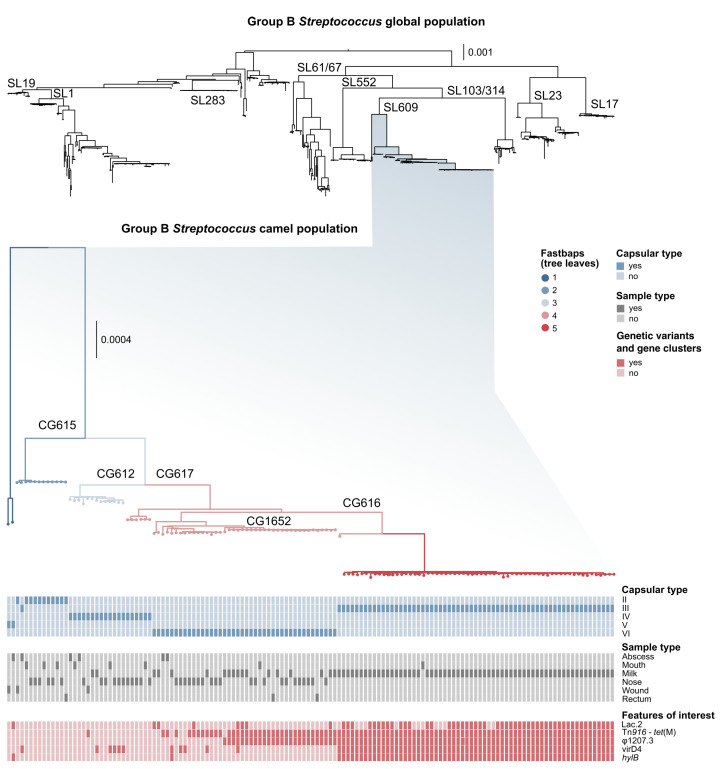
Midpoint-rooted trees of the global group B *Streptococcus* (GBS) population, and of the camel-specific sublineage (SL)609. All but one sequence type (ST)1 camel genome described to date belong to this SL. Clonal groups (CG) within SL609 are indicated on the tree branches. Leaves and branches are coloured based on fastbaps populations, as indicated by the legend. Coloured blocks below show capsular type, sample type, and selected features of interest based on Scoary results.

**Figure 2 pathogens-11-01025-f002:**
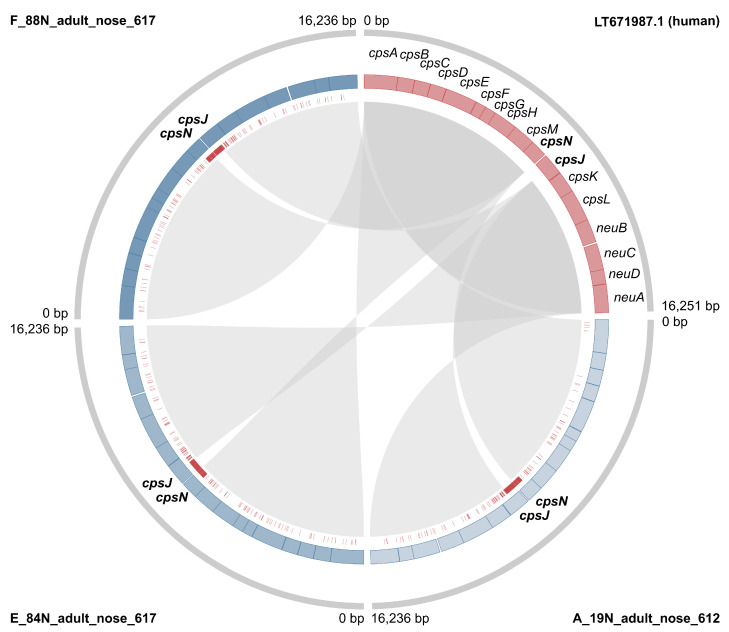
The capsular locus of the human reference genome for Group B *Streptococcus* capsular type IV is compared with three sequences that represent the overall phylogenetic diversity of type IV in GBS from camels (see Appendix A). External strips show the full *cps* locus (grey), internal strips the *cps* genes (pink: human, blue shades: camel), and the red bars indicate single nucleotide polymorphisms relative to the human GBS reference genome. Internal links show gene identity (default BLAST settings).

**Figure 3 pathogens-11-01025-f003:**
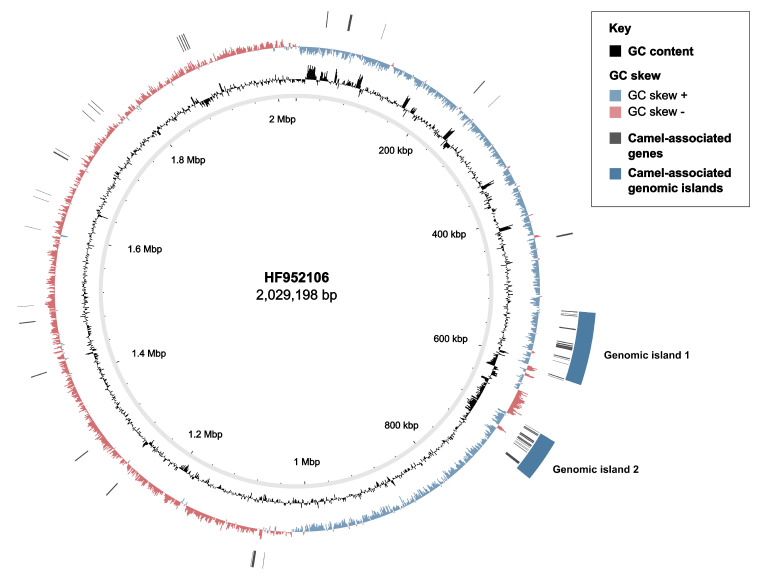
Circular genome map showing camel-associated accessory genome content in Group B *Streptococcus* based on Scoary, mapped to reference genome HF952106 (camel GBS, ST609, capsular type V). From inner to outer, circles show: position of the genome, GC content, GC skew, camel-associated genes (black blocks) and camel-associated genomic islands (GEI) (blue blocks). A lower GC content can be appreciated in the area corresponding to island 2 (GC island 2: 31% vs. GC whole genome: 35%) and in between the two GEI. Opposed GC skew, corresponding to the partial sequence of ICE*Sp*1108, can be observed in this area of the genome. Plot was generated with BRIG v0.95.

**Figure 4 pathogens-11-01025-f004:**
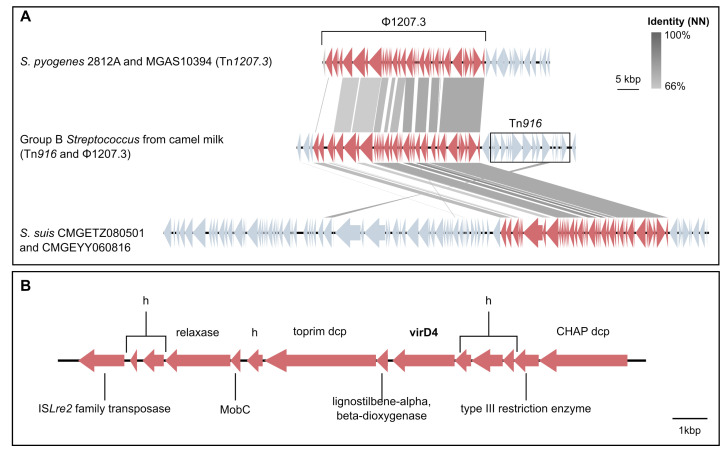
Genomic islands (GEI) that were associated with milk-derived Group B *Streptococcus* from camels upon within-host comparison with extra-mammary isolates: Tn*916*-ϕ1207.3 (**A**) and *virD4* GEI (**B**). (**A**) Visualisation of a BLASTn comparison between the Tn*916*-ϕ1207.3 associated with camel milk isolates (centre, from isolate 1M) and mobile elements from *Streptcoccus pyogenes* (top, Tn*1207.3 S. pyogenes* strain 2812A and MGAS10394) and *Streptococcus suis* (bottom, CMGETZ080501 and CMGEYY060816). Figure was obtained with Easyfig v2.2.2 [46] and modified with Inkscape v1.2.1 (www.inkscape.org). (**B**) Visualisation of gene orientation and annotation of the *virD4* GEI from camel GBS isolate 1M (see Appendix A). The contig on which this element was located ended at the right hand side of the CHAP domain containing protein (dcp). Hypothetical proteins are indicated with h.

## Data Availability

Genomic data for this study are available in publicly accessible repositories (either at the National Centre for Biotechnology Information or at the European Nucleotide Archive web pages, please consult the Supporting Material for accession numbers). New genomic data were made available in the European Nucleotide Archive, project number PRJEB53664. Data resulting from the analyses described in this paper are all contained within the article or the Appendix A.

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
