# Peer review of "How GBS Got Its Hump: Genomic Analysis of Group B Streptococcus from Camels Identifies Host Restriction as well as Mobile Genetic Elements Shared across Hosts and Pathogens"

_pathogens, 2022, doi:10.3390/pathogens11091025_

Round 1

Reviewer 1 Report

This manuscript showed evidence of camel-specific GBS distinct from other animal-GBS isolates. However, I have some comments appeared below.

1.      Line 19-25: The authors would like to show the important of GBS, however, GBS can cause infection in adults other than neonates. Please see Paveenkittiporn W, et al. Streptococcus agalactiae infections and clinical relevance in adults, Thailand. Diagn Microbiol Infect Dis. 2020;97(1):115005. It may be better if the author can mention this point.

2. A scheme (Flow chart) of the methods including sample origin should be shown. It will allow the readers to understand the process of study.

3. How many GBS you used in this study? Some from NCBI and some from your study. Please conclude the total.

4.  Line 174-176: What is the actual serotype of ST615 you mention? Please conclude it.

5.  Based on the Figure 1: it is interesting that serotype III of CG616 isolates were associated with milk (I assume “mastitis”), so, you can mention this in the result. Can you analyse statistics to show evidence of significance (p-value)? This will add your impact value.

6.  Subtopic of Markers of camel-association within GBS, were these genes significance? Statical analysis require to show, at least p-value. SO, the author can mention in Table.

7.  Subtopic “Markers of mastitis-association within camel GBS”. Based on Figure 1, Can you analyse statistics to show evidence of hylB and virD4 for its significance (p-value) or not? This will add your impact value.

8.  What is antimicrobial-resistance genes in camel-GBS? I think that it will be better if you can show.

9.   Please analyse “virulence genes” in camel-GBS. This can show in Table or picture.

Author Response

Response to Reviewer 1 Comments

Point 1: Line 19-25: The authors would like to show the important of GBS, however, GBS can cause infection in adults other than neonates. Please see Paveenkittiporn W, et al. Streptococcus agalactiae infections and clinical relevance in adults, Thailand. Diagn Microbiol Infect Dis. 2020;97(1):115005. It may be better if the author can mention this point.

Response 1: We thank the reviewer for highlighting this point. We have updated the introduction accordingly (lines 19-21).

Point 2: A scheme (Flow chart) of the methods including sample origin should be shown. It will allow the readers to understand the process of study.

Response 2: A flow chart with the methods has been added to the Supplementary material (as specified in lines 156-157).

Point 3: How many GBS you used in this study? Some from NCBI and some from your study. Please conclude the total.

Response 3: The total number of GBS genomes used for this study is 680, as stated in line 86. For clarity, this has also been added to the flow chart requested by the reviewer.

Point 4: Line 174-176: What is the actual serotype of ST615 you mention? Please conclude it.

Response 4: Based on the existing in silico typing methods, it is not possible to call a definitive serotype for this isolate. The discrepancies of the results between methods (Metcalf vs GBS-SBG) highlights the need for improved tools; in particular, there is a need for inclusion of animal genomes when developing genomic typing methods and WGS serotype definitions, as the authors highlight with the discrepancies of serotype IV (in silico result) and serotype Ia (in vitro results) (lines 190-195).

Point 5: Based on the Figure 1: it is interesting that serotype III of CG616 isolates were associated with milk (I assume “mastitis”), so, you can mention this in the result. Can you analyse statistics to show evidence of significance (p-value)? This will add your impact value.

Response 5: The association of CG/ST616 with milk/mastitis is described at lines 237-239 and 378-380. CG616 is known to be associated with milk/mastitis from a previous study (Seligsohn et al., 2021); this citation has been added to the text (line 239).

Genome-wide association studies (GWAS) are an unbiased statistical approach to detect genes that are positively or negatively associated with a trait (in this case: the host of origin and the type of sample, camel and camel milk). We have added paragraph to the introduction to clarify why we choose this approach most appropriate for our study (lines 77-83).

All genes reported and discussed in the Results and Discussion sections were significantly associated with camels or camel milk at p < 0.001. We have added a sentence to clarify this (lines 208-209). In addition, all exact p-values can be found in the Scoary output files that have been provided as supplementary material.

Point 6: Subtopic of Markers of camel-association within GBS, were these genes significance? Statical analysis require to show, at least p-value. SO, the author can mention in Table.

Response 6: Please, see the answer to the previous question.

Point 7: Subtopic “Markers of mastitis-association within camel GBS”. Based on Figure 1, Can you analyse statistics to show evidence of hylB and virD4 for its significance (p-value) or not? This will add your impact value.

Response 7: Please, see the answer to the previous question.

Point 8: What is antimicrobial-resistance genes in camel-GBS? I think that it will be better if you can show.

Response 8: Based on GWAS, the tet(M) gene for tetracycline resistance was identified as significantly and positively associated with mastitis in GBS from camels. This finding is consistent with previous work on geno- and phenotypic characterisation of GBS isolates from camel milk (Seligsohn et al., 2020 and 2021). A sentence to clarify this has been added to the manuscript (lines 240-241).

Point 9: Please analyse “virulence genes” in camel-GBS. This can show in Table or picture.

Response 9: Based on GWAS, one virulence gene , virD, was identified as statistically associated with camel GBS from mastitis isolates. Another virulence gene, vapE, was found in GBS from camel as part of a mobile genetic element (PICI4) that is specific to camels. Although those genes were known as virulence genes in other streptocococcal species, they had not previously been recognized as virulence genes in GBS, demonstrating the value of the unbiased nature of GWAS. These findings are discussed in lines 391-396 and lines 349-352, and the virulence genes are listed in the supplementary table with GWAS results.

Reviewer 2 Report

This is an excellent manuscript, presenting new data about the population structure of camel originin Streptococcus agalactiae. I support its publiction after appropriate minor modifications as outlined below:

The authors must used the recommended correct Microsoft Word template (https://www.mdpi.com/journal/pathogens/instructions)

Line 5: „Here, we describe...” – It is not scientifically sound, I would like to recommend to authors to avoid the using of personal verb forms throughout the manuscript (e.g.line 75 „we investigated”, etc.)

Line 5: „Camelus dromedarius” – in brackets, like in the line 53

Line 21: „differentiated” instead of „defined”

Lines 22-25: unclear sentence, please divide it

Line 57: different font

Line 59: „with mastitis, i.e. inflammation of the mammary gland.” – repetitive idea, or put „i.e.” in brackets

Line 81: „Dataset selection” – the subheadings must to be numbered

Line 82: it would be important for the reader to describe the selection strategy of the processed genomes (680 GBS genomes ?)

Lines 159, 223, 288, 317, 346, 367: it is not necessary the inserted empty lyne

Line 392: before the last paragraph, please insert „5. Conclusions” section, and highlight future research perspectives in the field.

Author Response

Response to Reviewer 2 Comments

Point 1: The authors must used the recommended correct Microsoft Word template (https://www.mdpi.com/journal/pathogens /instructions).

Response 1: The authors used the LaTex template (within Overleaf) provided by MDPI at https://www.mdpi.com/authors/latex.

Point 2: Line 5: „Here, we describe...” – It is not scientifically sound, I would like to recommend to authors to avoid the using of personal verb forms throughout the manuscript (e.g.line 75 „we investigated”, etc.)

Response 2: The authors believe this is more a matter of writing style than of scientific soundness. The usage of personal pronouns is becoming more common in scientific literature. We defer to the Editor for stylistic preferences.

Point 3: Line 5: „Camelus dromedarius” – in brackets, like in the line 53

Response 3: Changed accordingly.

Point 4: Line 21: „differentiated” instead of „defined”

Response 4: Changed accordingly.

Point 5: Lines 22-25: unclear sentence, please divide it

Response 5: We have changed the sentence to: “For example, hypervirulent clonal complex (CC) 17 is a human GBS clade specifically associated with neonates. In cattle, GBS is specifically associated with the mammary gland, where it causes mastitis.” (lines 22-24).

Point 6: Line 57: different font

Response 6: As the manuscript has been compiled within the LaTex template provided by MDPI, the font is the same throughout the text.

Point 7: Line 59: „with mastitis, i.e. inflammation of the mammary gland.” – repetitive idea, or put „i.e.” in brackets

Response 7: Sentence changed to: "In cattle, GBS is specifically associated with the mammary gland, where it causes infection and inflammation (mastitis)."

Point 8: Line 81: „Dataset selection” – the subheadings must to be numbered

Response 8: Changed accordingly.

Point 9: Line 82: it would be important for the reader to describe the selection strategy of the processed genomes (680 GBS genomes ?)

Response 9: We selected genomes to represent a wide range of geographical origins, host species, years of isolation and sequence types/serotypes. This improves on previous GWAS carried out on GBS, which only included genomes from four countries and with a statistical bias towards human isolates and genomes from outbreaks (Gori et al., 2020). All major GBS sublineages, as reported in the largest genomic population study on GBS so far (Richards et al., Mol Biol Evol, 2019), are included in our work. To avoid redundancy in the genomic data, which would have increased computational requirements without providing additional information on gene presence/absence, we selected one isolate per host/year/serotype/ST combination for each epidemiological unit, e.g. hospital or farm. We have updated the description of the dataset to explain this (lines 88-90) and provide details of all metadata in Supplementary file S1.

Point 10: Lines 159, 223, 288, 317, 346, 367: it is not necessary the inserted empty lyne

Response 10: Changed accordingly.

Point 11: Line 392: before the last paragraph, please insert „5. Conclusions” section, and highlight future research perspectives in the field.

Response 11: The “Conclusions” section is an optional one in MDPI journals (section 2.1.1. “Article” of the Author’s Guidelines https://www.mdpi.com/authors/layout).

We added a sentence highlighting as future work the importance of ongoing genomic surveillance of camel GBS to monitor the emergence of new and potentially resistant, virulent, or zoonotic strains (lines 402-404).

Round 2

Reviewer 1 Report

None